# Adverse Pregnancy Outcomes and Cardiovascular Disease: A Spanish Cohort

**DOI:** 10.3390/healthcare13070728

**Published:** 2025-03-25

**Authors:** Marta Miserachs, Cristina Martinez-Bueno, Almudena Castro, Vicente Pallarés-Carratalá, Antonia Pijuan-Domenech, Blanca Gordon, Alba Farràs, Ester Del Barco, Teresa Higueras, Elena Carreras, Maria Goya

**Affiliations:** 1Maternal-Foetal Medicine Unit, Department of Obstetrics, Vall d’Hebron Barcelona Hospital Campus, 08035 Barcelona, Spain; 2Sexual and Reproductive Health Services, Catalan Institute of Health, Barcelona University (UB), Gran Via de les Corts Catalanes, 587, 08007 Barcelona, Spain; 3Department of Cardiology, Hospital Universitario La Paz, 28046 Madrid, Spain; 4Health Surveillance Unit, Mutual Insurance Union, 12004 Castellon, Spain; 5Department of Medicine, Jaume I University, 12006 Castellon, Spain; 6Integrated Hospital Vall d’Hebron-Hospital Sant Pau Adult Congenital Heart Disease Unit, Vall d’Hebron Hospital Universitari, Vall d’Hebron Barcelona Hospital Campus, 08035 Barcelona, Spain; 7Department of Cardiology, Vall d’Hebron Hospital Universitari, Vall d’Hebron Barcelona Hospital Campus, CIBER-CV, 08035 Barcelona, Spain; 8Obstetrics and Gynecology Department, Universitat Autònoma de Barcelona, Plaça Cívica, 08193 Bellaterra, Spain

**Keywords:** women’s health, stroke, acute myocardial infarction, cardiovascular disease, coronary heart disease, pregnancy, preeclampsia, preterm birth, gestational diabetes, stillbirth, late miscarriage

## Abstract

**Background and Aims:** Emerging evidence suggests adverse pregnancy outcomes (APOs) may increase future cardiovascular risk. This study aimed to assess in a Spanish cohort the long-term risk of cardiovascular disease in women with APOs compared to those without such complications. **Methods:** A retrospective longitudinal cohort study was conducted at Hospital Vall d’Hebron (Barcelona, Spain), including pregnant women delivering between January 2010 and December 2015. Women with pre-existing medical conditions were excluded. APOs included preeclampsia, gestational diabetes, preterm birth, late miscarriage, and stillbirth. Cardiovascular events were defined as acute myocardial infarction or stroke. Both APO and non-APO groups were compared for their risk of cardiovascular events in the years following delivery, using unadjusted and adjusted models. **Results:** Out of 12,071 pregnant women delivered at Hospital Vall d’Hebron during the study period. 10,734 met the inclusion criteria (8234 in the non-APO group and 2500 in the APO group). The adjusted model revealed a significant association between APOs and cardiovascular events post-delivery (HR 2.5; 95% CI 1.4–4.4). Furthermore, an increased number of APOs (≥2) correlated with a higher risk of post-delivery cardiovascular events (HR 8.6; 95% CI 2.8–26.8). **Conclusions:** Women with adverse pregnancy outcomes (APOs), particularly those experiencing preeclampsia, preterm birth, and late miscarriage, exhibit an elevated long-term risk of cardiovascular events. Our findings highlight that these associations persist even after adjusting for traditional cardiovascular risk factors, indicating that APOs may independently influence long-term cardiovascular health. This underscores the importance of recognizing pregnancy as a critical window for early cardiovascular health interventions and counseling. Addressing these risks proactively could improve long-term health outcomes for women with a history of APOs.

## 1. Introduction

Cardiovascular disease (CVD) remains the leading cause of mortality in developed countries and is frequently underdiagnosed in women [1,2,3]. Historically, efforts to reduce CVD risk have predominantly targeted postmenopausal women, focusing on well-established risk factors such as diabetes, smoking, hypertension, and hypercholesterolemia. While these interventions have contributed to a decline in CVD mortality among women over 50, progress in reducing CVD risk in younger women remains limited [4,5].

Pregnancy induces significant physiological adaptations, including vascular and metabolic changes, designed to meet the increased energy demands of the mother and fetus [6]. Although these adaptations are necessary, they can unmask or exacerbate underlying conditions, potentially leading to adverse pregnancy outcomes (APOs) such as preeclampsia, gestational hypertension, preterm delivery, intrauterine fetal death, and small-for-gestational-age (SGA) infants [7,8,9,10,11,12,13,14,15,16,17].

Recent evidence suggests that APOs serve as markers of future cardiovascular risk, positioning pregnancy as a natural “stress test” that may reveal latent cardiovascular vulnerabilities [18]. Women who experience APOs are at greater risk of developing CVD later in life, highlighting pregnancy as a critical window for early intervention and risk stratification [19,20,21,22].

The interplay between APOs and long-term cardiovascular health underscores the importance of postpartum monitoring and screening [23]. Current guidelines recommend assessing cardiovascular risk factors in the postpartum period, though standardized protocols for follow-up remain lacking. Early identification of women at risk through lifestyle modifications, such as improved diet and physical activity, could mitigate the progression of CVD [24,25,26]. However, the long-term implications of APOs on cardiovascular health remain an area of active research.

Given these considerations, our study aims to investigate the association between APOs and the long-term development of cardiovascular disease in a Spanish cohort. We hypothesize that women with a history of APOs will demonstrate a higher incidence of CVD compared to those with uncomplicated pregnancies.

## 2. Methods

This was a retrospective longitudinal cohort study conducted on the reference area population of the Vall d’Hebron Barcelona Hospital Campus (Barcelona, Spain). Women were eligible for inclusion if delivery occurred from January 2010 to December 2015 at 12 to 42 weeks of gestation.

Patients with a history of diseases that increase cardiovascular risk, such as maternal cardiopathy, diabetes mellitus, or maternal chronic hypertension, were excluded.

Participants were divided into two groups depending on whether they had had an APO during pregnancy or not: the APO group and the non-APO group. The following outcomes during pregnancy were considered APOs: preeclampsia/gestational hypertension, gestational diabetes, preterm birth, late miscarriage, or stillbirth.

Diagnoses of preeclampsia or gestational hypertension were performed according to internationally recommended criteria. Gestational hypertension is generally defined as de novo hypertension during pregnancy (≥140 mmHg systolic and/or ≥90 mmHg diastolic), and preeclampsia is defined as new-onset high blood pressure after 20 weeks of gestation (≥140 mmHg systolic and/or ≥90 mmHg diastolic) or worsening of previous high blood pressure in addition to new-onset proteinuria (protein to creatinine ratio >300 or ≥1þ on the dipstick test) [27]. Fetal crown-rump length measurements determined gestational age during the first-trimester ultrasound or based on the date of the last menstrual period [28]. Preterm delivery was defined as delivery between 24 weeks and <37 weeks [29]. Gestational diabetes was diagnosed by the NDDG criteria [30,31]. Late miscarriage is a pregnancy loss that occurs during the second trimester of pregnancy; we have considered late miscarriages between 12 to 20 weeks of gestation [32]. Stillbirth was defined as the death or loss of a fetus at ≥20 weeks of gestation and just before or during birth [33].

Regarding cardiovascular events, acute myocardial infarction (AMI) was diagnosed according to the European Society of Cardiology guidelines, which include detecting an increase or decrease of a cardiac biomarker (with at least one value above the 99th percentile of the upper reference limit) and at least one of the following: symptoms of myocardial ischemia, new ischemic ECG changes, pathological Q waves on the ECG, imaging evidence of viable myocardium loss, or new regional wall motion abnormalities in a pattern consistent with an ischemic etiology or intracoronary thrombus detected on angiography or autopsy [34]. Stroke was defined as a neurological deficit attributed to an acute focal injury within the central nervous system of a vascular origin [35].

Chronic hypertension was defined as the persistent elevation of systolic blood pressure (SBP) to a level of ≥140 mmHg and/or diastolic blood pressure (DBP) to ≥90 mmHg in women aged 18 years or older, at which the benefits of treatment decidedly outweigh the associated risks [36].

Databases from the hospital and the primary care setting were searched to identify cases of a cardiovascular event years after delivery. AMI, stroke, or both were considered cardiovascular events.

Cases with CVD were recorded until December 2021, within 6 to 11 years after delivery, depending on the year of delivery. All cases with confirmed cardiovascular events were reviewed separately to identify classical risk factors for CVD.

This study was approved by the Hospital Vall d’Hebron Barcelona Ethics Committee (VHEC) (PR(AMI)541/2020) in December 2020.

Statistical analysis involved comparing the non-APO group with the APO group. Categorical data were reported as frequency and percentage. Comparisons between groups were estimated by the Mann-Whitney U, Kruskal-Wallis, chi-square, or Fisher exact tests, as appropriate. The statistical significance level was set at *p* < 0.05. A linear regression analysis was performed to adjust demographic factors (maternal age at birth, obesity (BMI ≥ 30), and smoking status) to compare the rate of cardiovascular events in both groups. The IBM SPSS Statistics v25 software was used for data analysis.

## 3. Results

During the period between 2010 and 2015, 12,071 pregnant women delivered at the Hospital Vall d’Hebron (Barcelona, Spain). Of these, 1377 women were excluded: 912 due to difficulty obtaining data and 425 due to previous medical history associated with a higher risk of cardiovascular events.

A total of 10,734 patients were included in the analysis. Two groups were created based on the presence of APOs. The non-APO group included 8234 women, while the APO group included 2500 women (Figure 1).

An analysis of the APO group was performed. A single APO was observed in 2236 women, and 2 or more APOs were observed in 264 women. The single most common APO was preterm birth (n = 1617, 72.3%), gestational diabetes (n = 280, 12.5%), preeclampsia (n = 187, 8.4%), late miscarriage (n = 79, 3.5%), and stillbirth (n = 81, 3.2%). Preterm birth and preeclampsia were the most common combination of APOs (n = 112, 4.24%) (Table 1). 

A comparative analysis of baseline characteristics between the two groups was conducted (Table 2).

Statistically significant differences in several key variables, including maternal age at birth, smoking status, and obesity (defined as a BMI ≥ 30), were observed when comparing the APO group and the non-APO group. Specifically, women within the APO group exhibited older maternal age at birth and a notably elevated prevalence of obesity and smoking compared to the non-APO group. Furthermore, in the evaluation of conception-related factors, our investigation unveiled significant distinctions between the groups.

Regarding perinatal outcomes, the APO group had a significantly higher rate of twin pregnancies (18% vs. 1.9%; *p* < 0.01) and lower birth weights (2550 g vs. 3290 g; *p* < 0.01) (Table 3).

The characteristics of women presenting cardiovascular events years after delivery were also evaluated for both groups. A total of 25 cardiovascular events occurred in the entire population of participants: 16 strokes and 9 AMIs. Of these, 12 events occurred in the APO group (0.5%, 12/2500) and 13 events occurred in the non-APO group (0.2%, 13/8234). None of the cardiovascular events led to maternal death (Table 4). For these patients with cardiovascular events years after delivery, the mean maternal age at the time of birth was significantly higher in the APO group than in the non-APO group (35 vs. 32, *p* < 0.001). In contrast, the presence of obesity (BMI ≥ 30) was more frequently observed in the non-APO group as compared with the APO group (15.4% vs. 0%, *p* < 0.01) (Table 5).

An evaluation of the APOs present in the group with cardiovascular events was performed. While no distinct pattern of APOs was discernible among women who had experienced an AMI, the cohort of women who had suffered a stroke exhibited a notable prevalence of preterm birth and the concurrent occurrence of preterm birth and preeclampsia (Table 6).

A potential correlation between the history of APOs and the subsequent development of cardiovascular events years after delivery was investigated (Table 7). It was observed that there was an average time of 5.8 years from the moment of childbirth to the onset of these cardiovascular events (Figure 2). The adjusted model by obesity (BMI ≥ 30), maternal age at birth, and smoking status showed a significant association between the presence of APOs during pregnancy and the development of cardiovascular events after pregnancy (HR 2.5; 95% CI 1.5–4.4). 

Specifically, conditions such as preeclampsia (HR 4.4; 95% CI 1.3–15.0), preterm birth (HR 2.9; 95% CI 1.3–6.6), and late miscarriage (HR 5.1 (1.2–22.2)) were particularly associated with cardiovascular events. Furthermore, our investigation revealed a markedly heightened risk of AMI in women who had experienced either stillbirth (HR 16.1; 95% CI 2.0–129.1) or late miscarriage (HR 13.6; 95% CI 1.5–121.9). It is important to note that this association remains unadjusted due to the limited number of cases available for analysis. There were no cardiovascular events in women with gestational diabetes, so it was impossible to establish a correlation between these APOs and future cardiovascular events.

In addition, the cumulative effect of experiencing multiple APOs on the risk of adverse cardiovascular events after pregnancy (Table 8) was evaluated. Our adjusted model, accounting for obesity (BMI ≥ 30), maternal age at delivery, and smoking status, demonstrated a progressive increase in the risk of any cardiovascular event with each additional APO experienced (HR 4.1; 95% CI 1.2–13.8).

Finally, the assessment of chronic hypertension risk revealed that women in the APO group exhibit a 66% higher likelihood of developing chronic hypertension than women in the non-APO group (Figure 3).

## 4. Discussion

### 4.1. Principal Findings

This is the first Spanish data population study on pregnancy and long-term maternal cardiovascular health that shows an association between APOs during pregnancy and the risk of cardiovascular events years after delivery.

The main finding of our study is that the presence of an APO is associated with an increased risk of future cardiovascular events. This association is significant for preeclampsia, preterm birth, and late miscarriage.

It is well-documented that approximately 30% of women may encounter APOs during the course of their pregnancies [37]. In the context of our study, it was observed that 23.3% of the women presented with at least one APO. Our findings underscore preterm birth as the main APO. However, it is imperative to note that our study did not differentiate between spontaneous and iatrogenic preterm deliveries. This lack of distinction may have contributed to the elevated incidence of preterm deliveries observed within the APO cohort.

While maternal age and obesity are well-established risk factors for CVD, their comparative impact on cardiovascular risk in the post-pregnancy period concerning APOs remains uncertain. Recent investigations have documented an association between maternal age and an elevated incidence of adverse pregnancy outcomes, including preeclampsia, gestational diabetes, stillbirth, and preterm birth [38,39,40,41,42]. However, it is noteworthy that age is a well-established risk factor for CVD [43,44]. In our study, we observed a higher prevalence of obesity among women within the APO group compared to those in the non-APO group. Nevertheless, among women who subsequently experienced cardiovascular events years after pregnancy, obesity was more prevalent in the non-APO group. Interestingly, despite the lower frequency of obesity in the APO group, our adjusted model demonstrated that patients with a history of APOs exhibited a heightened risk of future CVD. This compelling finding leads us to the conclusion that APOs exert a significant influence on the development of CVD, independent of traditional cardiovascular risk factors. Nonetheless, further studies are needed to determine whether cardiovascular risk in women with advanced maternal age may be attributed to having more adverse pregnancy outcomes or purely to age.

### 4.2. Results in the Context of What Is Known

Our findings align with existing literature that highlights an elevated risk of CVD among women who have experienced preeclampsia and preterm birth [15]. A significant 4-fold increased risk for any cardiovascular disease was observed in the present study for women presenting preeclampsia and almost a 3-fold risk for those presenting a preterm birth. The underlying mechanism for developing CVD due to adverse pregnancy complications remains unclear. It has been reported that CVD may develop due to placental and/or vascular dysfunction, causing an increased inflammatory state [45] and, consequently, an increased risk of CVD years after delivery. However, it remains to be seen whether APOs are involved in new pathophysiological processes leading to an increased long-term cardiovascular risk.

### 4.3. Clinical Implications

Growing evidence in recent years suggests that pregnancy offers an opportunity to identify women at a high risk of CVD [15,16,17,18,19] and subsequently to provide prevention strategies and monitoring. This presents an opportunity for implementing preventive strategies and ongoing monitoring. However, a substantial gap persists in postpartum follow-up, referred to as the ‘4th trimester’. Several factors contribute to this deficiency.

First, it is imperative to recognize that healthcare professionals are often unaware of the importance of obstetric history in assessing CVD. Consequently, this knowledge gap results in suboptimal health counseling and culminates in inadequate postpartum follow-up care. This situation is compounded by the prevailing lack of awareness among most women regarding their own cardiovascular risk. The fourth trimester, being a period when a woman’s primary focus is on her newborn, further exacerbates the challenge of seizing the opportunity to provide comprehensive follow-up and counseling for women at risk of developing CVD.

Moreover, current CVD risk assessment tools predominantly rely on classical risk factors such as hypertension, diabetes mellitus, hypercholesterolemia, age, and smoking status. For most women in the fourth trimester, employing these tools that do not account for gender-specific risk factors results in a calculated 10 year cardiovascular risk below the threshold defined by most CVD prevention guidelines, rendering them ineligible for preventive interventions. Recent guidelines from the European Society of Cardiology [46], the American Heart Association [39], and the American College of Obstetricians and Gynecologists [47,48] recommend, including APOs such as preterm birth or preeclampsia in the CVD risk assessment.

## 5. Research Implications

Further research is essential to develop gender-specific risk assessment tools that can precisely evaluate cardiovascular disease (CVD) risk and clarify how adverse pregnancy outcomes (APOs) independently contribute to increased CVD risk [49]. Research should explore the role of conception-related factors, such as assisted reproductive technologies and spontaneous conceptions, in the occurrence of APOs and their subsequent impact on cardiovascular health. Additionally, it is important to examine how different conception methods affect the vascular and metabolic adaptations during pregnancy, which may predispose women to long-term cardiovascular complications [50]. Such studies will provide deeper insights into the distinct pathways through which APOs affect cardiovascular health, helping tailor preventive and interventional strategies more effectively.

Understanding the pathophysiological links between APOs and long-term cardiovascular risks is crucial. This includes exploring the role of inflammatory pathways, endothelial dysfunction, and metabolic changes initiated during pregnancy, which may persist postpartum and contribute to cardiovascular disease development [51,52]. The association between common APOs such as preeclampsia, gestational diabetes, and subsequent cardiovascular events such as myocardial infarction and stroke needs detailed investigation to identify specific biomarkers and potential therapeutic targets [53,54].

Public health strategies to enhance postpartum follow-up during the fourth trimester should also be explored to enhance women’s long-term health prospects. This includes implementing comprehensive follow-up programs that integrate cardiovascular risk assessment with routine postpartum care, thereby addressing the gap in care for women with a history of APOs [49]. Such strategies could involve multidisciplinary teams to ensure that cardiovascular risk factors are adequately managed and monitored following delivery, potentially reducing long-term morbidity and mortality associated with cardiovascular disease [55].

Efforts to develop and validate gender-specific risk assessment tools that incorporate factors such as age, obesity, and history of APOs could lead to more precise risk stratification and more personalized preventive care strategies. These tools would help clinicians identify at-risk women at an earlier stage and implement interventions that could substantially alter the trajectory of their cardiovascular health [2].

Emerging evidence links adverse pregnancy outcomes to future cardiovascular risks, with vitamin D deficiency identified as a key shared factor. Research, such as the stratified randomized trial by Rostami et al. [56], shows that screening and treating vitamin D deficiency during pregnancy can significantly improve outcomes; 900 participants in the control group had a mean 25(OH)D concentration of 11 ng/mL, and those in the treatment group also had 11 ng/mL at baseline but were supplemented to increase 25(OH)D over 20 ng/mL. Furthermore, McDonnell et al. demonstrated that adequate maternal 25(OH)D levels (benefit above 20 ng/mL) reduce preterm birth risks by 60%, suggesting a substantial cardiovascular benefit postpartum [57]. These findings emphasize the importance of monitoring vitamin D levels during pregnancy to mitigate immediate and long-term health risks, and it would be interesting to address this issue in this population. A recent study also shows that a very low 25(OH)D concentration could be considered a risk factor for both adverse pregnancy outcomes and CVD [58].

### Strengths and Limitations

Our study has certain potential limitations that deserve consideration. First, we performed a retrospective analysis of the data, which precluded the inclusion of some demographic factors known to influence the development of cardiovascular disease (CVD), such as ethnicity and socioeconomic status. Furthermore, despite including a substantial cohort of pregnant women in our study, the incidence of events, particularly AMI, remained relatively low. This limited occurrence of events challenges our ability to definitively establish a causal relationship between any of the evaluated APOs and the subsequent risk of AMI. Consequently, it is prudent to interpret our findings as exploratory and hypothesis-generating.

To provide more conclusive insights into this matter, future research should consider undertaking multicentric studies featuring larger participant cohorts. Such studies would facilitate the accrual of a more substantial number of cardiovascular events for comprehensive analysis.

## 6. Conclusions

Women with APOs, particularly those who experience preeclampsia, preterm birth, and late miscarriage, face a significantly elevated long-term risk of cardiovascular events. These findings reinforce the notion that pregnancy itself acts not only as a critical phase for immediate maternal and fetal care but also as a pivotal opportunity for initiating comprehensive cardiovascular health counseling. By identifying APOs as independent risk factors for cardiovascular disease, we emphasize the need for targeted interventions during pregnancy that can potentially mitigate long-term health risks. This approach highlights the dual benefit of prenatal care, ensuring both immediate and future health security for mothers.

## Figures and Tables

**Figure 1 healthcare-13-00728-f001:**
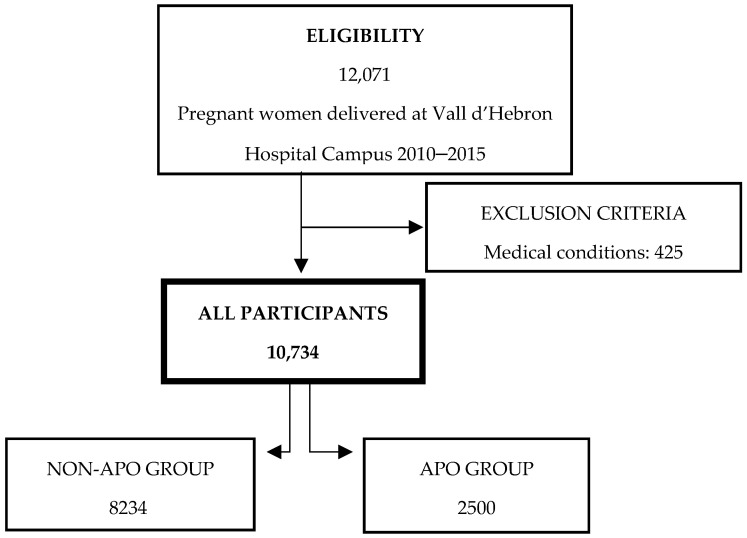
Flowchart of the pregnant women included in the study‰.

**Figure 2 healthcare-13-00728-f002:**
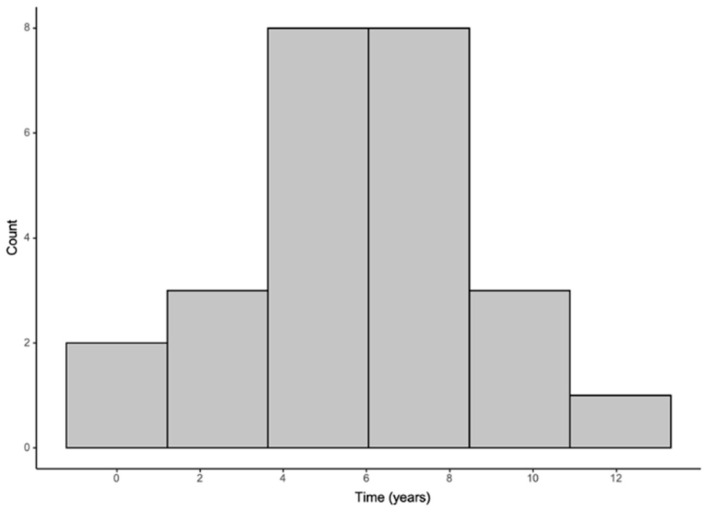
Time to event among women with cardiovascular events.

**Figure 3 healthcare-13-00728-f003:**
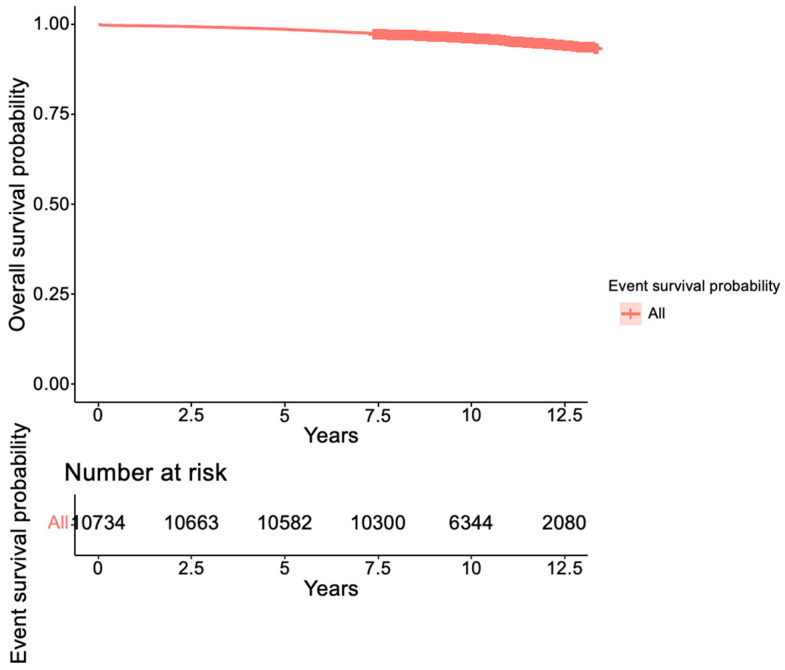
Survival probability for developing chronic hypertension.

**Table 1 healthcare-13-00728-t001:** Characteristics of the APO cohort.

	APO Cohort n = 2500
Preterm birth	1617 (64.7%)
Gestational diabetes	280 (11.2%)
Preeclampsia	187 (7.5%)
Previous late miscarriage	79 (3.1%)
Stillbirth	40 (1.6%)
Preterm birth + preeclampsia	112 (4.5%)
Previous late miscarriage + preterm birth	44 (1.8%)
Preeclampsia + gestational diabetes	12 (0.5%)
Preterm birth + gestational diabetes	12 (0.5%)
Previous late miscarriage + gestational diabetes	7 (0.3%)
Stillbirth + preeclampsia	2 (0.1%)
Stillbirth + gestational diabetes	2 (0.1%)
Preterm birth + gestational diabetes + preeclampsia	5 (0.2%)
Previous late miscarriage + preterm birth + gestational diabetes	4 (0.2%)
Stillbirth + preterm birth	6 (0.2%)
Late miscarriage + preeclampsia	3 (0.1%)
Late miscarriage + preterm birth + preeclampsia	2 (0.1%)
Late miscarriage + preeclampsia + gestational diabetes	2 (0.1%)

**Table 2 healthcare-13-00728-t002:** Demographic characteristics for all participants versus occurrence of APO during pregnancy.

	Non-APO Cohort n = 8234	APO Cohort n = 2500	*p*-Value
Maternal age at birth (y)	31.2 (5.9)	32.7 (5.9)	<0.001
Obesity (BMI ≥ 30)	768 (9.3)	328 (13.1)	<0.0001
Smoking during pregnancy			0.006
No	6273 (76.2)	1826 (73.0)	
Yes	1066 (12.9)	372 (14.9)	
No data	895 (10.9)	302 (12.1)	
Conception			<0.001
Spontaneous	7028 (85.4)	1968 (78.7)	
Assisted	244 (3.1)	239 (9.6)	
No data	951 (11.5)	426 (10.6)	

**Table 3 healthcare-13-00728-t003:** Perinatal outcomes for all participants versus occurrence of APO during pregnancy.

	Non-APO Cohort n = 8234	APO Cohort n = 2500	*p*-Value
Twin pregnancies			<0.001
Yes	137 (1.9)	403 (18)	
No	7149 (98.1)	1834 (82)	
No data	948 (11.5)	263 (10.5)	
Pregnancy outcome			<0.001
Live birth	8208 (99.7)	2385 (95.4)	
Stillbirth	0 (0)	34 (2.4)	
Neonatal death	26 (0.3%)	32 (1.3%)	
Weight at birth (g)	3290 (480)	2550 (876)	<0.001
Gestational age at birth (weeks)	39.7 (1.2)	35.5 (4.1)	<0.001

**Table 4 healthcare-13-00728-t004:** Cardiovascular events in all participants.

	Stroke (n = 16)	Acute Myocardial Infarction (n = 9)
APO group	9/2500	3/2500
non-APO group	7/8234	6/8234

**Table 5 healthcare-13-00728-t005:** Demographic characteristics of women with cardiovascular events versus occurrence of APO during pregnancy.

	Cardiovascular Events in the Non-APO Cohort n = 13	Cardiovascular Events in the APO Cohort n = 12	*p*-Value
Maternal age at birth (y)	32 (7)	35 (5)	<0.001
Obesity (BMI > 30)	2 (15%)	0 (0%)	<0.001
Smoking during pregnancy			0.006
No	7 (54%)	9 (75%)	
Yes	4 (31%)	2 (17%)	
No data	2 (15%)	1 (8%)	
Conception			<0.001
Assisted	0 (0%)	0 (0%)	
Spontaneous	11 (85%)	10 (83%)	
No data	2 (15%)	2 (16%)	

**Table 6 healthcare-13-00728-t006:** Assessment of APOs in women presenting cardiovascular events.

a	
**APOs**	**Stroke (n = 16)**
Preterm birth	6 (37.5%)
Preterm birth + preeclampsia	2 (12.5%)
Late miscarriage	1 (6.25%)
**APOs**	**Acute myocardial infarction (n = 9)**
Stillbirth	1 (1.11%)
Preterm birth + late miscarriage	1 (1.11%)
Preterm birth + preeclampsia	1 (1.11%)

**Table 7 healthcare-13-00728-t007:** Correlation between APOs during pregnancy and adverse cardiovascular events years after delivery.

	Acute Myocardial Infarction	Stroke	Any Cardiovascular Event (Stroke OR Acute Myocardial Infarction)
Preterm birth	1.4 (0.3–6.6)	4.4 (1.7–11.9)	2.9 (1.3–6.6)
Preeclampsia	4.0 (0.5–31.8)	4.2 (0.9–19.0)	4.4 (1.3–15.0)
Late miscarriage	13.6 (1.5–121.9)	3.1 (0.4–24.0)	5.1 (1.1–22.2)
Stillbirth	16.1 (2.0–129.1)	--	5.6 (0.8–41.6)
Any APO	2.3 (0.9–5.8)	2.6 (1.3–4.9)	2.5 (1.5–4.4)

**Table 8 healthcare-13-00728-t008:** Correlation between the additive effect of more than one APO and adverse cardiovascular events years after delivery.

	Acute Myocardial Infarction	Stroke	Any Cardiovascular Event (Stroke OR Acute Myocardial Infarction)
1 APO	0.6 (0.1–5.1)	3.3 (1.1–9.5)	2.1 (0.8–5.0)
≥2 APO	10.3 (2.1–51.2)	7.1 (1.4-34.9)	4.1 (1.2–13.8)

## Data Availability

The data used in this study are not publicly available due to ethical and privacy restrictions. However, anonymised data may be made available to the corresponding author upon reasonable request, subject to institutional and ethical approval.

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
