# Peer review of "Adverse Pregnancy Outcomes and Cardiovascular Disease: A Spanish Cohort"

_healthcare, 2025, doi:10.3390/healthcare13070728_

Round 1
Reviewer 1 Report (Previous Reviewer 3)
Comments and Suggestions for Authors
I congratulate the authors for their work! The authors revised the manuscript:
- The introduction is comprehensive and provide a backgorund about the topic
- Disscussion supports the results
- Disscussion section was revised and research implications was added
- A minor suggestion: the refferences section need to be revised.
Author Response
Thank you for your positive feedback and constructive suggestions.
We have already revised the references section to ensure its accuracy and completeness. This update has been included in the latest version of the manuscript.
Thank you again for your attention to detail and support.
Reviewer 2 Report (Previous Reviewer 2)
Comments and Suggestions for Authors
|
merging evidence links adverse pregnancy outcomes to future cardiovascular risks, |
337 |
|
with vitamin D deficiency identified as a key shared factor. Research, such as the stratified |
338 |
|
randomised trial by Rostami et al.(X), shows that screening and treating vitamin D defi- |
339 |
|
ciency during pregnancy can significantly improve outcomes. Furthermore, McDonnell et |
340 |
|
al. demonstrated that adequate maternal 25(OH)D levels (above 40 ng/mL) reduce pre- |
341 |
|
term birth risks by 60%, suggesting a substantial cardiovascular benefit postpartum (XX). |
Comment: A good point but it could be improved.
First, the Rostami et al. article should be described. The important thing was that 900 participants in the control group had a mean 25(OH)D concentration of 11 ng/mL, and that those in the treatment group also had 11 ng/mL at baseline but were supplemented to increase 25OHD over 20 ng/mL. The McDonnell et al. paper found that most of the benefits were above 20 ng/mL, so not important to mention 40 ng/mL.
A recent article should also be cited:
Zhang N, Wang Y, Li W, Wang Y, Zhang H, Xu D et al. Association between serum vitamin D level and cardiovascular disease in Chinese patients with type 2 diabetes mellitus: a cross-sectional study. Sci Rep 2025; 15(1): 6454. e-pub ahead of print 20250222; doi: 10.1038/s41598-025-90785-8
It found that risk of CVD was significantly increased for participants with 25(OH)D below 12 ng/mL. Thus, it meshes with the Rostami et al. article that very low 25(OH)D concentration is a risk factor for both adverse pregnancy outcomes and CVD.
Author Response
Thank you for your insightful comments.
We have updated our manuscript to include a detailed description of the Rostami et al. study and clarified the significant health benefits above 20 ng/mL of 25OHD as noted in the McDonnell et al. paper. Additionally, we have cited the recent study by Zhang N et al. to strengthen the discussion on vitamin D levels and cardiovascular risks.
These changes have been highlighted in yellow in the revised manuscript for your review.
Thank you again for your valuable feedback.
This manuscript is a resubmission of an earlier submission. The following is a list of the peer review reports and author responses from that submission.
Round 1
Reviewer 1 Report
Comments and Suggestions for Authors
The manuscript ”ADVERSE PREGNANCY OUTCOMES AND CARDIOVAS- 2 CULAR DISEASE: A SPANISH COHORT” by Miserachs M et al. is very interesting and essential in the field of cardiovascular diseases, especially when we are speaking about pregnant women and their CVD risk. The study was well conducted, the results support the study's title, and the findings are important, especially for clinicians. The methodology used reflected the main results. The manuscript was well written, without scientific flaws or self-citation.
I recommend it for publication, but it is unsuitable for Nutrients and its current issue.
Author Response
Dear reviewer,
Thank you for your thoughtful and positive feedback on our manuscript, "ADVERSE PREGNANCY OUTCOMES AND CARDIOVASCULAR DISEASE: A SPANISH COHORT." We greatly appreciate your recognition of the importance of our study and your kind words regarding the quality of our research and writing.
We understand your assessment regarding the manuscript's suitability for the current Nutrients issue. However, we hope there may be an opportunity for the paper to be considered for publication in a new issue very soon.
Once again, thank you for your valuable time and consideration.
Reviewer 2 Report
Comments and Suggestions for Authors
Out of 12,071 pregnant women delivered at Hospital Vall d'Heb- 34 ron during the study period. 10,734 met the inclusion criteria (8,234 in the non-APO group and 2,500 35 in the APO group).
Women with APOs, especially those experiencing preeclampsia, preterm 39 birth, and late miscarriage, exhibited an elevated long-term risk of cardiovascular events. These 40 findings suggest that pregnancy could serve as an opportunity to initiate counselling on cardiovas- 41 cular health.
Comment: 2500/12,071 = 20.7%, which seems high, especially since it appears that most of the APO can be prevented.
To the conclusion could be added that pregnant women should be counseled at the first prenatal visit with an MD that they should have their serum 25(OH)D concentration measured and if it is below 50 nmol/L, should take vitamin D supplements to increase it above 50 nmol/L or, better, above 75 to 100 nmol/L using bolus doses with retesting after a month or so. See:
Maternal 25(OH)D concentrations 40 ng/mL associated with 60% lower preterm birth risk among general obstetrical patients at an urban medical center.
McDonnell SL, Baggerly KA, Baggerly CA, Aliano JL, French CB, Baggerly LL, Ebeling MD, Rittenberg CS, Goodier CG, Mateus Niño JF, Wineland RJ, Newman RB, Hollis BW, Wagner CL.PLoS One. 2017 Jul 24;12(7):e0180483. doi: 10.1371/journal.pone.0180483.
Effectiveness of Prenatal Vitamin D Deficiency Screening and Treatment Program: A Stratified Randomized Field Trial.
Rostami M, Tehrani FR, Simbar M, Bidhendi Yarandi R, Minooee S, Hollis BW, Hosseinpanah F.J Clin Endocrinol Metab. 2018 Aug 1;103(8):2936-2948. doi: 10.1210/jc.2018-00109.
Dietary advise might also be given.
Also, it would be useful to state what the guidelines for pregnancy are in Spain, and what doctors discuss at the first prenatal visit.
Looking at the reference list, it appears that only 3 peer-reviewed journal articles from 2021 were cited but none published after 2021. I consider this unacceptable. I suggest searching Google Scholar for more recent papers on topics discussed in this manuscript.
Significant digits. The general rule is that no more non-zero digits should be given than are justified by the uncertainty of the value.
See "Too many digits: the presentation of numerical data"
https://www.ncbi.nlm.nih.gov/pmc/articles/PMC4483789/
If the uncertainty (or difference when comparing numbers) is greater than about 7%, only two non-zero digits are justified.
P values should be given to two decimal places unless the first two are 00 or the number lies between 0.045 and 0.054. If the first two are 00, then only one non-zero digit can be given.
Thus,
|
0.00572 |
Should be
0.006
|
32.4 (7.0) |
Should be
32 (7)
|
2 (15.4%) |
Should be
2 (15%) since N is less than 150
Please review all numbers in abstract, text, tables, and figures and adjust accordingly.
Author Response
Thank you very much for your insightful comments and recommendations regarding our manuscript. We have thoroughly reviewed and revised the manuscript, making sure all your suggestions have been implemented across the suggested sections. We respond to your comments in the attached document, with our responses highlighted in blue for ease of reference.

Reviewer 3 Report
Comments and Suggestions for Authors
The article aimed to study the long-term correlations between cardiovascular events and adverse pregnancy outcomes.
The introduction section gives an overview of the issue. The methodology is clearly presented; statistical analysis is robust; the results are well presented and supported by statistical analysis; conclusions are consistent with the results. The iconography is very helpful. It is easier to follow the methodology and the results.
It's an interesting and well-written article and worth publishing.
Minor revisions:
- the introduction section could be furter devoloped (cardiovascular risk in pregnancy, vascular and metabolic changes in pregnancy, what is known about adverse pregnancy outcomes and cardiovascular health)
- the size of figure 1 could be smaller
- more comments about the differences between adverse pregnancy ouctomes group and non-adverse pregnancy outcomes group about the evaluation of conception related factors
- discussing the possible pathophysiology underlying the association between adverse pregnancy outcomes and cardiovascular events
Author Response
Thank you for your detailed feedback. We have addressed your comments, making the necessary revisions to our manuscript. Each change has been highlighted in blue in the attached document for easy reference. We hope these adjustments meet your approval and look forward to your further suggestions.

Round 2
Reviewer 1 Report
Comments and Suggestions for Authors
The article is substantially improved and prepared to be accepted.
Author Response
Thank you for your feedback and for acknowledging the improvements made to the manuscript.
Reviewer 2 Report
Comments and Suggestions for Authors
Emerging evidence suggests adverse pregnancy out- 24 comes (APOs) may increase future cardiovascular risk.
Comment: Actually, adverse pregnancy outcomes and cardiovascular risk have shared risk factors. One of the more important ones is vitamin D deficiency. Thus, the authors should provide a short discussion on this point. This is an excellent opportunity to provide useful information to the medical system in Spain and elsewhere.
These papers must be cited:
Effectiveness of Prenatal Vitamin D Deficiency Screening and Treatment Program: A Stratified Randomized Field Trial.
Rostami M, Tehrani FR, Simbar M, Bidhendi Yarandi R, Minooee S, Hollis BW, Hosseinpanah F.J Clin Endocrinol Metab. 2018 Aug 1;103(8):2936-2948. doi: 10.1210/jc.2018-00109.
Maternal 25(OH)D concentrations 40 ng/mL associated with 60% lower preterm birth risk among general obstetrical patients at an urban medical center.
McDonnell SL, Baggerly KA, Baggerly CA, Aliano JL, French CB, Baggerly LL, Ebeling MD, Rittenberg CS, Goodier CG, Mateus Niño JF, Wineland RJ, Newman RB, Hollis BW, Wagner CL.PLoS One. 2017 Jul 24;12(7):e0180483. doi: 10.1371/journal.pone.0180483.
There are many good papers regarding vitamin D and CVD
Vitamin D Deficiency Increases Mortality Risk in the UK Biobank : A Nonlinear Mendelian Randomization Study.
Sutherland JP, Zhou A, Hyppönen E.Ann Intern Med. 2022 Nov;175(11):1552-1559. doi: 10.7326/M21-3324. Epub 2022 Oct 25.
The Effects of Vitamin D Supplementation and 25-Hydroxyvitamin D Levels on the Risk of Myocardial Infarction and Mortality.
Acharya P, Dalia T, Ranka S, Sethi P, Oni OA, Safarova MS, Parashara D, Gupta K, Barua RS.J Endocr Soc. 2021 Jul 15;5(10):bvab124. doi: 10.1210/jendso/bvab124.
Vitamin D and cardiovascular diseases: Causality.
Wimalawansa SJ.J Steroid Biochem Mol Biol. 2018 Jan;175:29-43. doi: 10.1016/j.jsbmb.2016.12.016.
Iurciuc M, Buleu F, Iurciuc S, Petre I, Popa D, Moleriu RD, Bordianu A, Suciu O, Tasdemir R, Dragomir R-E, et al. Effect of Vitamin D Deficiency on Arterial Stiffness in Pregnant Women with Preeclampsia and Pregnancy-Induced Hypertension and Implications for Fetal Development. Biomedicines. 2024; 12(7):1595. https://doi.org/10.3390/biomedicines12071595
Links between Vitamin D Deficiency and Cardiovascular Diseases.
Mozos I, Marginean O.Biomed Res Int. 2015;2015:109275. doi: 10.1155/2015/109275.
I know that medical doctors like to see good results from randomized controlled trials. Unfortunately, most vitamin D RCTs were poorly designed, conducted, and analyzed, and cannot be used for policy recommendations. Thus, observational studies such as the above should be used.
Comparing the Evidence from Observational Studies and Randomized Controlled Trials for Nonskeletal Health Effects of Vitamin D.
Grant WB, Boucher BJ, Al Anouti F, Pilz S.Nutrients. 2022 Sep 15;14(18):3811. doi: 10.3390/nu14183811.PMID: 36145186
Critical Appraisal of Large Vitamin D Randomized Controlled Trials.
Pilz S, Trummer C, Theiler-Schwetz V, Grübler MR, Verheyen ND, Odler B, Karras SN, Zittermann A, März W.Nutrients. 2022 Jan 12;14(2):303. doi: 10.3390/nu14020303.
Significant digits. The general rule is that no more non-zero digits should be given than are justified by the uncertainty of the value.
See "Too many digits: the presentation of numerical data"
https://www.ncbi.nlm.nih.gov/pmc/articles/PMC4483789/
If the uncertainty (or difference when comparing numbers) is greater than about 7%, only two non-zero digits are justified.
P values should be given to two decimal places unless the first two are 00 or the number lies between 0.045 and 0.054. If the first two are 00, then only one non-zero digit can be given.
Thus, (HR 8.6; 95% CI 2.76- 37 26.82). should be (HR 8.6; 95% CI 2.8- 37 26.8).
|
2 APO |
10.33 (2.09-51.19) |
7.09 (1.44-34.86) |
4.14 (1.24-13.81) |
Should be
|
2 APO |
10.3 (2.1-51.2) |
7.1 (1.4-34.9) |
4.1 (1.2-13.8) |
|
Preterm birth |
1.36 (0.28-6.57) |
Should be
|
Preterm birth |
1.3(0.3-6.6) |
Please review all numbers in abstract, text, tables, and figures and adjust accordingly.
Author Response
Thank you for your constructive feedback and valuable suggestions. We have incorporated all the recommended changes into the manuscript. We believe these modifications have greatly enhanced the clarity and quality of our paper. We appreciate your guidance and support throughout this revision process.
